# Cognitive and Emotional Symptoms Induced by Chronic Stress Are Regulated by EGR1 in a Subpopulation of Hippocampal Pyramidal Neurons

**DOI:** 10.3390/ijms24043833

**Published:** 2023-02-14

**Authors:** Anna Sancho-Balsells, Sara Borràs-Pernas, Verónica Brito, Jordi Alberch, Jean-Antoine Girault, Albert Giralt

**Affiliations:** 1Departament de Biomedicina, Facultat de Medicina, Institut de Neurociències, Universitat de Barcelona, Casanova 143, 08036 Barcelona, Spain; 2Institut d’Investigacions Biomèdiques August Pi i Sunyer (IDIBAPS), 08036 Barcelona, Spain; 3Centro de Investigación Biomédica en Red Sobre Enfermedades Neurodegenerativas (CIBERNED), 08036 Barcelona, Spain; 4Production and Validation Center of Advanced Therapies (Creatio), Faculty of Medicine and Health Science, University of Barcelona, 08036 Barcelona, Spain; 5Inserm UMR-S 1270, 75005 Paris, France; 6Science and Engineering Faculty, Sorbonne Université, 75005 Paris, France; 7Institut du Fer à Moulin, 75005 Paris, France

**Keywords:** dendritic spines, hippocampus, engram, memory, behavioral despair

## Abstract

Chronic stress is a core risk factor for developing a myriad of neurological disorders, including major depression. The chronicity of such stress can lead to adaptive responses or, on the contrary, to psychological maladaptation. The hippocampus is one of the most affected brain regions displaying functional changes in chronic stress. Egr1, a transcription factor involved in synaptic plasticity, is a key molecule regulating hippocampal function, but its role in stress-induced sequels has been poorly addressed. Emotional and cognitive symptoms were induced in mice by using the chronic unpredictable mild stress (CUMS) protocol. We used inducible double-mutant Egr1-CreERT2 x R26RCE mice to map the formation of Egr1-dependent activated cells. Results show that short- (2 days) or long-term (28 days) stress protocols in mice induce activation or deactivation, respectively, of hippocampal CA1 neural ensembles in an Egr1-activity-dependent fashion, together with an associated dendritic spine pathology. In-depth characterization of these neural ensembles revealed a deep-to-superficial switch in terms of Egr1-dependent activation of CA1 pyramidal neurons. To specifically manipulate deep and superficial pyramidal neurons of the hippocampus, we then used Chrna7-Cre (to express Cre in deep neurons) and Calb1-Cre mice (to express Cre in superficial neurons). We found that specific manipulation of superficial but not deep pyramidal neurons of the CA1 resulted in the amelioration of depressive-like behaviors and the restoration of cognitive impairments induced by chronic stress. In summary, Egr1 might be a core molecule driving the activation/deactivation of hippocampal neuronal subpopulations underlying stress-induced alterations involving emotional and cognitive sequels.

## 1. Introduction

One of the most-studied and important environmental risk factors associated with the development of neurological disorders, such as schizophrenia, major depressive disorder (MDD) and other cognitive problems, is chronic stress [1,2,3,4]. In response to acute stressful events, an organism responds with adaptive changes on many levels (psychological, behavioral and physiological) which help to cope with the situation [5]. In healthy individuals, this response is limited in time. However, if this response is prolonged in time and disproportionate in salience, it leads to significant disorders such as anxiety and depression [6].

In recent years, many neuroimaging-based studies have shown that the hippocampus is one of the brain regions most affected in psychiatric disorders such as MDD. Specifically, they show a significant reduction in hippocampal volume [7,8], and similar results have been observed in animal models of depression [9,10]. Furthermore, chronic stress has been related to the impairment of hippocampal-dependent memory [11], although studies about associated microstructural changes, such as spine loss, have been less consistent [12,13]. Interestingly, these alterations may depend on the hippocampal subregion [14], the stressor type and its duration [15,16] and the sex [16,17] of the animals, among other things. On the other hand, the rich cellular diversity recently described in the hippocampus [18] increases the complexity of the cellular interactions in the progression of stress-induced depression. It has been proposed that neurons of the CA1 region of the hippocampus can be classified as deep and superficial pyramidal cells along the radial axis. Deep and superficial pyramidal neurons have distinct transcriptional profiles and different electrophysiological properties [19,20]. These previous reports suggest that both subtypes could play different roles in the hippocampus in both normal and stressful conditions.

Growing evidence suggests that immediate early genes (IEGs) can be altered in different neuropsychiatric conditions that are associated with deficits in neuronal activity and plasticity, including major depression [21,22]. Early growth response 1 (*Egr1*) is a classical IEG and a zinc finger transcription factor that modulates the expression of many genes involved in crucial neuronal functions, such as neurotransmission, synaptic plasticity, learning and response to stress [22]. As with other IEGs, *Egr1* expression can be induced by a variety of signals, such as injury or stress [21]. Moreover, Egr1 is a very dynamic transcription factor, being upregulated or downregulated in the brain depending on the chronicity of the stress protocol [23,24]. From these previous studies, we hypothesize that *Egr1* expression could be tightly regulated depending on the chronicity of stress and be a potential molecular event underlying the cognitive and emotional symptoms induced by chronic stress.

In the present work, we mapped in mice the Egr1-dependent activation of neural ensembles in several brain regions at different time points in a chronic unpredictable mild stress protocol (CUMS). We identified Egr1-activity-tagged neurons in CA1 that increase or decrease depending on the chronicity of the stress protocol. Furthermore, the cellular identity of these activated cells changed along the CUMS protocol, shifting from deep to superficial pyramidal neurons. Finally, we show that manipulations of superficial or deep pyramidal cells play distinct roles in the modulation of the sequelae induced by chronic stress.

## 2. Results

### 2.1. Egr1-Dependent Neural Activation in Hippocampal CA1 Depends on the Chronicity of the Stress

It is widely accepted that several brain regions, such as the amygdala, cingulate cortex, septum and hippocampus, among others, play a role in chronic stress [25,26,27,28], and that persistence of stressful periods can determine the progression and severity of associated depressive symptoms [29]. Here, we mapped the forebrain by counting the density of GFP-positive cells in brain regions highly affected by chronic stress. Since previous studies suggest that Egr1 is upregulated in short periods of stress [30] whereas the same protein is downregulated in long periods of stress [11], we hypothesized that this protein could be an excellent candidate to map and manipulate the activation/deactivation of neural ensembles dependently on the chronicity of the stressful events. Thus, we took advantage of Egr1-CreERT2 BAC transgenic mice [31]. These mice express Cre recombinase fused to modified estrogen receptor (ERT2) only activated by 4-hydroxytamoxifen (4-HT), under the control of the *Egr1* promoter. When these mice are injected with saline, no recombination is observed [31,32]. To identify which neural cells are activated/deactivated in short-term vs. long-term stress conditions, we crossed Egr1-CreERT2 mice with R26RCE mice, a reporter line in which EGFP expression requires recombination by Cre (Figure 1A). We then subjected Egr1-CreERT2xR26RCE double transgenic mice to a CUMS protocol. Mice were separated into three groups: control non-stressed (CNT group), short-term-stressed (2 days, STS group), and long-term-stressed (28 days, LTS group). Each group received one injection of 4-HT 30 min before the presentation of a stressful stimulus during the last two consecutive days of the CUMS protocol (Figure 1B). First, we observed that the density of GFP-positive cells in the septum (Figure 1C), the amygdala (Figure 1D) and the cingulate cortex (Figure 1E) was similar between groups. The LTS group showed an increase in GFP-positive cells in the paraventricular nucleus compared to the CNT and STS groups (Figure 1F). In the striatum we could not detect significant differences between any condition (Figure 1G). In the hippocampal CA3 region (Figure 1H), GFP-positive cells were reduced in the LTS group compared with CNT group. Finally, in the CA1 we observed remarkable bidirectional changes in the number of GFP-positive cells depending on the chronicity of the stress protocol (Figure 1I). Concretely, GFP-positive cells were increased in the STS group but decreased in the LTS group when compared with the CNT group. From the latter result, we hypothesize that the CA1 could be highly sensitive to the chronicity of the stress. Finally, we performed and immunofluorescence to validate that all the GFP-positive cells also expressed Egr1 (Figure 1J).

### 2.2. Specific Spine Density Alterations in Egr1-Positive Activated Cells

Since we observed that stress induced opposite changes in the activation of Egr1-dependent neural ensembles in the CA1 depending on its duration, we hypothesized that such changes could be accompanied by alterations in synapse numbers in these particular ensembles. To explore this, we used Egr1-CreERT2 mice (Figure 2A) injected in the dorsal CA1 with AAV-CaMKII-FLEX-EGFP to allow the quantification of dendritic spines specifically in the Egr1-positive ensembles (Figure 2B). Again, we used a control group (CNT), a group of mice subjected to stress for 2 days (STS), and a group of mice subjected to stress for 28 days (LTS) (Figure 2C). Apical dendrites from CA1 pyramidal neurons were analyzed in the three groups (Figure 2D). We observed that spine density in apical dendrites was significantly decreased in the LTS group whereas it was not changed in the STS mice (Figure 2E). Taken together, the results highlighted that, although short-term stress induced a similar trend, only long-term stress resulted in significant changes in spine density from activated pyramidal neurons of the CA1. To evaluate whether such changes in spine density were global or specific to the Egr1-dependent ensembles, we then evaluated spine density in pyramidal neurons of the dorsal CA1 by using the DioListic method in different groups of wild type mice subjected to 28 days of CUMS and compared them with unstressed control mice (Figure 2G). The results showed that the density of dendritic spines was similar between control and LTS mice when it was counted in pyramidal neurons in an unspecific manner. Although the methods employed in each experiment are different, these data suggest that not all pyramidal cells present alterations in spine density upon CUMS but mainly those that are activated in a Egr1-dependent manner. 

### 2.3. Different Hippocampal Subpopulations Are Activated Depending on Stress Duration

We observed that the CA1 displayed the highest contrast in neural activation depending on the duration of the stress protocol, as indicated by Egr1-activity induction. Such activation was accompanied by time-dependent changes in dendritic spine densities. This led us to hypothesize that a possible progressive cellular reorganization (in terms of Egr1-dependent activation) could take place in the CA1 during the progression of chronic stress. To characterize such potential cellular reorganization, we immunostained the CA1 of Egr1-CreERT2xR26RCE mice in the three conditions (CNT, STS and LTS) for different markers, including GFAP, a marker of astrocytes, PV, a marker of parvalbumin interneurons, and MAP2, a general marker for CA1 pyramidal cells. We first observed that neither GFAP (Figure 3A,B) nor PV (Figure 3A,D) colocalized with GFP (Egr1-dependent activated neural cells) in any group. In contrast, almost all the GFP-positive cells colocalized with MAP2-positive cells in the three groups (Figure 3A,C). We also immunostained hippocampal sections of the three groups of mice for Calb1, which identifies the superficial pyramidal neurons of CA1, whereas pyramidal neurons negative for Calb1 are considered to be deep pyramidal cells [19]. Remarkably, our quantifications revealed that the percentage of GFP-positive CA1 pyramidal cells co-stained with Calb1 was different in the three groups (Figure 3E). The percentage of double-stained Calb1-positive/GFP-positive cells was low in the CNT group, significantly increased in the STS group, and reached its highest level in the LTS group (Figure 3E). However, these percentages did not reflect changes in the total number of GFP-positive cells depending on the duration of the CUMS protocol (see Figure 1I). Therefore, we also analyzed the total number of Calb1-positive/GFP-positive neurons, corresponding to activated superficial pyramidal neurons. This number was increased in the STS group, whereas in the LTS group the number of Calb1-positive/GFP-double-positive neurons was similar to in the CNT group (Figure 3F). In contrast, the density of Calb1-negative/GFP-positive neurons (activated deep pyramidal neurons) was unchanged in the STS group compared to the CNT group but significantly decreased in the LTS group (Figure 3G). To confirm that these alterations were not due to global changes in total Calb1 levels, we measured Calb1 levels using western blotting (Figure 3H,I) and immunofluorescence (Figure 3J,K). Our quantifications (Figure 3H,J) confirmed that total Calb1 levels are unaltered by stress. All these results taken together indicate that, although there is a progressive reduction of CA1 GFP-positive pyramidal neurons at the end of the CUMS protocol (Figure 1I), there is a transient and concomitant increase in the percentage of Egr1-dependent activated neurons, which are in turn Calb1-positive neurons (superficial pyramidal neurons). In contrast, Calb1-negative neurons (deep pyramidal neurons) are severely and progressively deactivated (in terms of Egr1-dependent activation) at the end of the CUMS protocol.

### 2.4. Chemogenetic Activation of Dorsal CA1 Deep Pyramidal Cells Does Not Prevent Sequelae Induced by CUMS

We observed that chronic stress induced a reduction in the number of Egr-1-dependent activated deep pyramidal cells in the CA1 (Figure 3G). To test whether such Egr1-dependent deactivation was relevant for the sequelae induced by chronic stress we aimed to activate them by using designer-receptors-exclusively-activated-by-designer-drugs technology (DREADDs). To selectively express hM3Dq DREADDs in deep pyramidal neurons we used Chrna7-Cre mice (Figure 4A). Chrna7-mice were injected in the dorsal hippocampus with an AAV vector construct containing a double-floxed inverted open-reading frame (DIO) sequence encoding hM3Dq-mCherry (Figure 4B). With such a design, only those Chrna7-positive neurons (deep pyramidal neurons) expressing Cre-recombinase would be able to recombine and express the DREADD in the correct orientation and only when CNO is given to the animal (in drinking water) would those cells become activated. We then subjected the mice to CUMS and divided them into four different groups: non-stressed treated with vehicle (CNT VEH group), non-stressed treated with CNO (CNT CNO), long-term (28 days)-stressed treated with VEH (LTS VEH) and long-term-stressed (28 days) treated with CNO (LTS CNO) (Figure 4C). In parallel, we used a small subgroup of mice treated with CNO for 2 days but without behavioral manipulation to avoid unspecific changes due to extensive experimentation. In these mice, immunofluorescence staining confirmed that only deep pyramidal cells were transduced and that CNO in drinking water induced the activation of Chrna7-positive neurons (cFos+ cells) (Figure 4D–F). We then analyzed the % of cFos+ cells out of those transduced and clearly observed that treatment with CNO induced activation of those transduced neurons (Figure 4E). Regarding the mice subjected to the CUMS procedure, we only used mice with correct allocation of the viral transduction in the CA1 (Figure 4F). Interestingly, we first observed that both groups of LTS mice presented increased locomotor activity in the open field test (Figure 4G). Next, to assess cognitive sequelae after CUMS, we performed the novel object location test, which is a task sensitive to detecting cognitive impairment upon CUMS [11,33,34]. In this test, only the CNT VEH and CNT CNO groups were capable of discriminating between the new and the old locations of the object (Figure 4H). Finally, we analyzed the time spent struggling in the forced-swim test. Both groups of LTS mice spent less time struggling specifically during the first minutes of the test compared to both groups of CNT mice (Figure 4I,J). To exclude the possibility of unspecific or off-target effects of CNO per se, we repeated the same experimental design as in Figure 4C without surgery. Hence, control mice received normal housing or the CUMS protocol while they were treated with VEH or CNO in drinking water. CNO did not produce any effect on locomotion in the open field when comparing the vehicle and CNO conditions (Figure 4K). Similarly, we did not find any effect mediated by CNO either in the novel object location task (Figure 4L) or in the forced-swim test (Figure 4M,N). These results ruled out unspecific off-target effects induced by CNO in the behavioral tasks performed. Altogether, our data suggest that the activation of Chrna7-positive (deep) neurons during the CUMS protocol has no effect on the CUMS-induced sequelae.

### 2.5. Specific Downregulation of Egr1 in CA1 Superficial Pyramidal Cells during CUMS Improves Depressive Sequelae

Trying to upregulate the activity of the deep pyramidal cells by using DREADDs was not efficient in the prevention of the sequelae induced by stress. Next, we decided to prevent effects mediated by the early increase in Egr1-dependent activated cells in the superficial pyramidal cells (Figure 3E–G) by reducing their endogenous Egr1 levels. We hypothesized that suppressing the increase in Egr1 in superficial neurons found at the beginning of the stress response could prevent some alterations induced by CUMS. To specifically modulate superficial neurons, we used Calb1-cre mice (Figure 5A). We thus transduced only superficial CA1 pyramidal (Calb1-positive) cells with shRNA against *Egr1* (Figure 5B) or with a GFP CNT virus. Mice were then divided into four different conditions: CNT mice injected with control AAV (AAV-CNT), CNT mice injected with an AAV-shRNA against *Egr1* (AAV-shEGR1), chronically stressed mice injected with control AAV (AAV-CNT+CUMS) and chronically stressed mice injected with an AAV-shRNA against *Egr1* (AAV-shEGR1+CUMS). After the CUMS protocol, mice were tested for different behavioral tasks as illustrated in our experimental design (Figure 5*C*), and, finally, they were sacrificed for viral transduction verification. We first showed widespread viral transduction in the CA1 superficial pyramidal neurons of the dorsal hippocampus (Figure 5D–F). Egr1 levels were specifically downregulated in those superficial pyramidal cells of the CA1 that were transduced with the SH-viral vector (Figure 5E). Then, in the open field test, both groups of mice subjected to the CUMS presented hyperlocomotion compared to the CNT groups (Figure 5G). We then used the NOLT to test spatial memory. AAV-CNT mice were able to distinguish between the old and new object positions. Conversely, AAV-shEGR1 mice could not differentiate the new location of the object. Interestingly, the deficits in spatial memory caused by CUMS (as observed in the AAV-CNT+CUMS mice) were alleviated in the AAV-shEGR1+CUMS mice (Figure 5H). We then subjected the mice to the forced-swim test to evaluate behavioral despair. In the first two minutes of the test (Figure 5I), both groups of mice subjected to CUMS significantly spent less time swimming compared to non-stressed mice. This was maintained for the last four minutes of the task (Figure 5J). Interestingly, in the last four minutes of the test, we observed that stressed mice injected with SH- against *Egr1* swam significantly more than stressed mice injected with the control AAV (Figure 5J). Finally, we also verified that Calb1 levels in superficial CA1 neurons were not affected by our experimental conditions (Appendix A). In summary, inhibition of *Egr1* expression in Calb1-positive (superficial) neurons corrected different sequelae caused by CUMS. It is noteworthy that since in this experiment we used AAV5-CAG-FLEX-GFP as a control instead of a scramble sequence the results should be taken with caution.

## 3. Discussion

In the present study, we identified changes in Egr1-dependent activated cells by CUMS in several brain regions. In the CA1 of the hippocampus, Egr1-dependent activated cells were of neuronal identity and differently affected by short- and long-term stress. Secondly, we found that those Egr1-dependent neurons activated during stress display dendritic spine alterations, and that these changes also depend on the duration of the CUMS protocol. We then determined the identity of the activated neurons and demonstrated that there is a change in the activated CA1 subpopulations, from being mostly Calb1-negative (deep) in non-stressed mice to being Calb1-positive (superficial) in long-term-stressed mice. We then used chemogenetic and genetic approaches to directly modulate these two hippocampal subpopulations. We found that inhibiting Egr1 expression in superficial pyramidal (Calb1-positive) neurons but not activating deep pyramidal neurons (Calb1-negative) alleviated some of the CUMS-induced sequelae.

Previous literature has reported effects on several brain regions in terms of neural activity in stress-related disorders [26,27]. Here, using mice expressing inducible Cre recombinase under the promoter of the immediate early gene *Egr1*, we describe changes in different brain areas depending on the chronicity of the CUMS protocol. First, long-term-stressed (LTS) mice presented an increased number of Egr1-dependent activated cells in the paraventricular nucleus compared to CNT mice, confirming previous literature [28]. In contrast, in the CA3, we found fewer Egr1-dependent activated cells in the LTS group. Interestingly, in the CA1 of the hippocampus we found opposite differences in the numbers of active neurons depending on stress duration. Egr1-dependent activated cells were increased after a short period of stress, while they were decreased after long periods of stress. In line with this, other studies have also shown dynamic changes in Egr1 using different stress paradigms, including social defeat, exposure to predators and chronic mild stress [21,22]. Moreover, some groups have found increased levels of Egr1 in the hippocampus upon acute stress [24,35,36,37,38,39,40,41] whereas other groups found decreased Egr1 levels when the stress was continuous in time ([42,43,44], but see [45,46,47]). Thus, our work reinforces the idea that chronicity is a core characteristic of stress, which determines if Egr1 is going to be up- or downregulated. Altogether, this indicates that Egr1 is very dynamic and sensitive to stress duration [46,48,49] and places the CA1 as a core brain region that differentially reacts depending on stress chronicity.

Concomitant with a progressive decrease in Egr1-dependent activated CA1 pyramidal cells we found specific alterations in dendritic spine density in the same cells, whereas the same parameter was not observed when using general methods to analyze spine density in neurons. It is widely accepted that stress induces changes in spine density, and that these changes depend on the brain region studied and the type of stress presented [13]. Here, we observed that when spine density is only measured in those significantly Egr1-dependent activated neurons, relevant alterations are found. Supporting this idea, although stress-induced dendritic spine pathology is evident in CA3 pyramidal cells [50,51,52,53,54,55], it is much less clear in CA1 pyramidal cells. Some authors have shown reduced spine density [46,55,56,57] or unchanged spine density [51,52,53,58] or even increased spine density [54,59,60] in CA1 pyramidal neurons in different stress paradigms. Based on our results and previous literature, we conclude that changes in dendritic spines in CA1 pyramidal neurons upon chronic stress could depend, at least, on two dimensions: first, they could depend on the duration of the stress, and, second, these changes in spine density could be limited to different neuronal subpopulations. Thus, our results may shed light on the discrepancies in the literature regarding changes in spine density in the CA1 upon stress. Finally, a potential mechanism by which increased Egr1 activity may lead to decreased spine density in these particular neuronal ensembles could be by reducing PSD-95 expression and inducing AMPAR endocytosis, as previously described [61].

We also observed that the type of hippocampal neuron activated is different depending on stress duration. Concretely, Egr1-dependent activated cells were preferentially deep CA1 pyramidal cells in control conditions, whereas Egr1-dependent activated cells were preferentially CA1 superficial pyramidal cells in chronic stress conditions. Our results also indicate that the modulation of Calb1-positive (superficial) cells per se has an impact on the pathophysiology of stress, since decreasing Egr1 levels in only this subpopulation produced an improvement in behavioral despair and spatial memory loss. Supporting this idea, it has been shown that Calb1 levels increase in CA1 pyramidal neurons upon stress [62]. Furthermore, reducing Calb1 levels in the CA1 is enough to alleviate memory loss in stress models [63]. Our results probably also indicate that, with only few days of stress, molecular Egr1-dependent events in superficial pyramidals are initiated and all the changes that we observed in 28 days of stress could be just a consequence triggered by those initial changes. This reinforces the idea that preventing this initial Egr1 increase in superficial cells would prevent both spatial learning deficits and behavioral despair induced by CUMS. In line with this, since Egr1 is a critical factor in encoding the enduring behavioral effects of stress in the hippocampus [64,65,66] we speculate that, possibly, the increase in Egr1 at the beginning of the stress induces the depressive-like symptoms whereas the Egr1 downregulation at the end of stress would induce the deficiencies in spatial learning. If our hypothesis is true, the fact that inhibiting Egr1 at the beginning of CUMS improves both FST and NOLT would indicate that the late effects (NOLT deficiencies and decreased Egr1-positive engrams) are dependent on the first effects (increase in Egr1-positive engrams and emotional sequelae). Finally, we also conclude that appropriate levels of this transcription factor in the superficial CA1 pyramidal neurons are mandatory for proper hippocampal function, as described elsewhere [67,68,69], since aberrant increases (induced by stress) or decreases in Egr1 (induced by hippocampal delivery of shRNA against *Egr1* in naïve mice) can both provoke deleterious effects on hippocampal-dependent memory.

## 4. Materials and Methods

### 4.1. Animals

For this study, we used Egr1-CreER^T2^ mice [31]. These mice carry a bacterial artificial chromosome (BAC) including the Egr1 gene in which the coding sequence was replaced by that of the CreERT2 fusion protein. Egr1-CreER^T2^ mice were crossed with R26RCE mice (Gt(ROSA)26Sortm1.1(CAG-EGFP)Fsh/Mmjax, Strain 004077, The Jackson Laboratory. RRID:IMSR_JAX:004077), which harbor the R26R CAG-boosted EGFP (RCE) reporter allele with a loxP-flanked STOP cassette upstream of the enhanced green fluorescent protein (EGFP) gene, to create double-heterozygous-mutant Egr1-CreER^T2^ × R26RCE mice. We also used Chrna7-cre mice (Tg(Chrna7-cre)NP348Gsat/Mmucd, GENSAT) and Calb1-cre mice (Calb1-T2A-dgCre-D, Jackson Labs, #023531. RRID:IMSR_JAX:023531). All the animals employed were males. Genotypes were determined from an ear biopsy as described elsewhere [70] and following manufacturer’s instructions. All mice were maintained in a C57BL/6 background strain and housed together in numerical birth order in groups of mixed genotypes (3–5 mice per cage). The animals were housed with access to food and water ad libitum in a colony room kept at 19–22 °C and 40–60% humidity, under an inverted 12:12 h light/dark cycle (from 08:00 to 20:00). All animal procedures were approved by local committees [Universitat de Barcelona, CEEA (10141) and Generalitat de Catalunya (DAAM 315/18)], in accordance with the European Communities Council Directive (86/609/EU).

For the experiments shown in Figure 1 and Figure 3, 7 adult (4-month-old) male mice were used per group (total n = 21 mice). For the experiments shown in Figure 2A–F, 7 adult (6–7-month-old) male mice were used per group (total n= 21 mice). For the experiments shown in Figure 2G–I, 3–4-month old mice were employed. For the experiments shown in Figure 4A–G,I,J, 60 adult (5–6-month-old) mice were used (CNT VEH n= 13; CNT CNO n = 16; LTS VEH n = 15 and LTS CNO n = 16). In Figure 4H, adult (5–6 month-old) mice were used (CNT VEH = 18; CNT CNO n = 16; LTS VEH n = 15 and LTS CNO n = 16). For the experiments shown in Figure 4K–N, 50 male adult mice (5–6-month-old) were used (CNT VEH n = 12; CNT CNO n = 12; LTS VEH n = 13 and LTS CNO n= 13). For the experiments shown in Figure 5, 52 male adult mice (7–8-month-old) were used (CNT GFP n = 10; CNT SH n = 13; LTS GFP n = 14 and LTS SH n = 15).

### 4.2. Drugs

4-hydroxytamoxifen (H6278, Sigma-Aldrich, St. Louis, MO, USA) was dissolved in 1 mL ethanol (100%). Corn oil (C8267, Sigma-Aldrich) was then added, and ethanol was evaporated by incubating the open tubes at 50 °C overnight in the dark. The volume of corn oil was adjusted for injecting the dose of active drug at 50 mg/kg.

Clozapine N-oxide (BML-NS105-0025, Enzo) was administered via drinking water as previously described [71]. The required CNO stock solution was then added to 1% sucrose water to reach a final concentration of 1 mg/kg/day. Water was changed 3 times per week.

Trimethoprim (Santa Cruz, sc-237332, CA, USA) was reconstituted in water at a concentration of 30 mg/100 mL. After surgery, all mice received TMP in drinking water for one week to induce Cre recombinase activity specifically in Calb1-cre mice.

### 4.3. Stereotaxic Surgery and Viral Transduction In Vivo

Animals were stereotaxically injected with one of the following adeno-associated viruses (AAVs): pAAV5-hSyn-DIO-hM3D(Gq)-mCherry (Addgene, #44361. RRID:Addgene_44361), AAV5-CAG/EF1a-DIO-mCherry-mEGR1-shRNAmir (#shAAV-258146, Vector Biolabs) and AAV5-CAG-FLEX-GFP (UNC, Vector Core). Following anesthesia with isoflurane (2% induction, 1.5% maintenance) and 2% oxygen, mice were bilaterally injected with AAVs (3 × 10^12^ GS per injection) in the CA1 of the hippocampus. We used the following coordinates (millimeters) from bregma: antero-posterior, −2.0; lateral, ±1.5; and dorso-ventral, −1.3. AAV injection was carried out in 4 min. The needle was left in place for 2 min for complete virus diffusion. After recovery, mice were returned to their home cage for 3 weeks. 

### 4.4. Chronic Unpredictable Mild Stress (CUMS)

One day before the beginning of the CUMS procedure described elsewhere [72], mice were individually housed and maintained in isolation for the entire experiment. The CUMS procedure (Table 1) followed a random weekly schedule of commonly used mild stressors (one per day) including restraint, food or water deprivation, home cage inclination and others. The CUMS protocol lasted 28 days for the long-term-stressed (LTS) group. Short-term-stressed (STS) mice were housed individually for one day prior to the stress procedure and received two days of stress (forced swim and restraint). Both the STS and LTS groups received the same stressors during the last two days (when they received 4-HT) in order to make them comparable. CNT mice remained undisturbed in their home cages for all the procedure.

### 4.5. Behavioral Tests

Open field (OF) and novel object location test (NOL): For the novel object location test (NOL), an open-top arena (40 × 40 × 40 cm) with visual cues placed in the walls of the apparatus was used. Mice were first habituated to the arena (1 day, 15 min). We considered this first exposition to the open arena as an open field paradigm. We monitored total travelled distance (in cm) as a measure of locomotor activity. On day 2, two identical objects (A1 and a2) were placed on one side of the arena and mice were allowed to explore them for 10 min. Exploration was considered when the mouse was in contact with object and sniffed it. 24 h later (D3), one object was moved from its original location to the diagonally opposite corner and mice were allowed to explore the arena and the objects for 5 min. At the end of each trial, defecations were removed, and the apparatus was cleaned with 30% ethanol. Animal tracking and recording were performed using the automated SMART junior 3.0 software (Panlab, Spain).

Forced swim test: The forced-swim test was used to evaluate behavioral despair. Animals were subjected to a 6 min trial during which they are forced to swim in an acrylic glass (35 cm height × 20 cm diameters) filled with water, and from which they cannot escape. The time that the test animal spent in the cylinder without making any movements except those required to keep its head above water was measured.

All the tests were conducted during the light cycle and all mice were randomized throughout the day. Only one test was conducted per day. 

### 4.6. Tissue Fixation, Immunofluorescence and Confocal Imaging

Animals were deeply anaesthetized and subsequently intracardially perfused with 4% (weight/vol) paraformaldehyde in 0.1 M phosphate buffer. Brains were dissected out and kept for 48 h in 4% paraformaldehyde 0.1 M phosphate buffer in agitation. After fixation, free-floating coronal sections (40 µm) were obtained using a vibratome (VT1000. Leica Microsystems CMS GmbH, Mannheim, Germany).

For immunofluorescence, sections were first washed twice in PBS and incubated in 50 mM NH4Cl for 30 min. Blocking and permeabilization were performed for 1 h in PBS-T with 0.02% azide, 3% NGS and 0.2% BSA. Primary antibodies were diluted in blocking solution, and sections were incubated overnight at 4 °C in agitation. The following primary antibodies were used: MAP2 (1:500; #M1406. RRID:AB_477171. Sigma-Aldrich, St. Louis, MO, USA), GFP FITC-conjugated (1:500, #Ab6662. RRID:AB_305635. Abcam, Cambridge, UK), Egr1 (1:1000, #4154S, Cell Signalling, Danvers, Massachusetts, USA), cFos (1:150, #sc-52. RRID:AB_2106783. Santa Cruz Biotechnology, Dallas, TX, USA), GFAP (1:500, Dako #z0334. RRID:AB_10013382) and Parvalbumin (1:1250; Sigma #P3088. RRID:AB_477329). Secondary antibodies (Cy3- or Cy2-coupled fluorescent secondary antibodies, 1:200, Jackson ImmunoResearch Laboratories catalogue #715-165-150, RRID:AB_2340813 and #715-545-150, RRID:AB_2340846, respectively) were diluted in blocking solution, and tissues were incubated for 2 h at room temperature. Nuclei were stained for 10 min with 4′,6-diamidino-2-phenylindole (DAPI; catalogue #D9542, Sigma-Aldrich). The sections were mounted onto slides and cover-slipped with Mowiol. Images (at 1024 × 1024-pixel resolution) in a mosaic format were acquired with a Leica Confocal SP5 with 63× or 40× oil-immersion objectives and standard (1 Airy disc) pinhole (1 AU), and frame averaging (3 frames per z step) were held constant throughout the study.

### 4.7. Immunobloting

Animals (n = 9–10 per group) were sacrificed by cervical dislocation. The hippocampus was dissected out, frozen using CO2 pellets, and stored at −80 °C until use. Briefly, the tissue was lysed by sonication in 250 μL of lysis buffer (50 mM Tris base (pH 7.5), 10 mM EDTA and 1% Triton X−100, supplemented with 1 mM sodium orthovanadate, 1 mM phenylmethylsulphonyl fluoride, 1 mg/mL leupeptin, and 1 mg/mL aprotinin). After lysis, samples were centrifuged at 15,000× *g* for 20 min. Supernatant proteins (15 mg) from total brain region extracts were loaded in SDS–PAGE and transferred to nitrocellulose membranes (GE Healthcare, LC, UK; Calle Gobelas 35- 37, 28023, Madrid, Spain). Membranes were blocked in TBS-T (150 mM NaCl, 20 mM Tris- HCl, pH 7.5, 0.5 mL Tween 20) with 5% BSA and 5% non-fat dry milk. Immunoblots were incubated overnight at 4° with anti-calbindin (Swant CB38) at 1:2000 in PBS with 0.2% Sodium Azide. After three washes in TBS-T, blots were incubated for 1 h at room temperature with anti-rabbit horseradish peroxidase-conjugated secondary antibody (1:2000; Promega, Madison, WI, USA) and washed again with TBS-T. Immunoreactive bands were visualized using the Western Blotting Luminol Reagent (Santa Cruz Biotechnology) and quantified by a computer-assisted densitometer (Gel-Pro Analyzer, version 4, Media Cybernetics, Rockville, MD, USA). For loading control, a mouse monoclonal antibody for tubulin was used (1:50,000, Sigma 083M4847V).

### 4.8. Gene Gun

Hippocampal neurons were labeled using the Helios Gene Gun System (Bio-Rad) as previously described [73]. Briefly, dye-coated particles were delivered by shooting into 200-μm coronal brain sections (obtained in a vibratome, Leica VT100), pointing at the hippocampus, at 80 psi through a membrane filter of 3 μm pore size and 8 × 10 pores/cm^2^ (Millipore, Burlington, MA, USA). Sections were stored at room temperature in PBS for 3 h protected from light and then subjected DAPI (#D1306, Invitrogen, Waltham, MA, USA) staining and mounted in Mowiol to be analyzed. 

### 4.9. Imaging Analysis

Cell counting was performed in area estimates for striatum, dorsal hippocampal CA1, dorsal hippocampal CA3, septum, para-ventricular nucleus, cingulate cortex, and amygdala and was performed in double-blind counting fashion relative to the different conditions using the ImageJ software (ImageJ 2.3.0/ 1.53q; Java 1.8). Images (at 512 × 512-pixel resolution) in a mosaic format were acquired with a Leica Confocal SP5 with a 20× normal objective and standard (1 Airy disc) pinhole (1 AU) and frame averaging (3 frames) were held constant throughout the study. No z-stacks were taken. To determine the neuronal subpopulation densities, we used the online Gaidi atlas as a reference, and we counted GFP-positive cells in coronal sections spaced 240 µm apart. The numbers of GFP-positive cells were relativized per area (500 µm^2^). For identification of the activated neurons shown in Figure 3, three 40-micron-thick slices per mouse separated 240 microns from each other were used to quantify these percentages. Percentages (%) of GFP-positive cells colocalizing with different markers (MAP2, GFAP, parvalbumin and calbindin1) were estimated per area as described above. For calbindin1 intensity quantification, 20 µm coronal sections were counted with a step size of 5 µm. Briefly, the ROI of the CA1 was delimited using the DAPI channel, and this ROI was applied in the calbindin1 channel, in which mean intensity was determined. Spine density counting was performed as previously described [73] using the ImageJ software. GFP- or DiI-labeled pyramidal neurons from CA1 of the dorsal hippocampus were imaged using a Leica Confocal SP5 with a ×63 oil-immersion objective (digital zoom 5×). Conditions such as pinhole size (1 AU) and frame averaging (4 frames per z-step) were held constant throughout the study. Confocal z-stacks were taken with a digital zoom of 5 and a z-step of 0.2 μm at 1024 × 1024 pixel resolution, yielding an image with pixel dimensions of 49.25 × 49.25 μm. For spine density analysis, we examined second-order dendrites. Thus, ramifications from the apical dendrite were analyzed. On average, 30 μm ± 2 of length was evaluated per dendrite, and the ratio was calculated as the number of dendrites/total length analyzed. A total of 17–27 dendrites from 7 different mice per group were quantified. For double-positive cell quantification, the criterion was to count as double-positive those cells presenting yellow pixels indicating colocalization of the marker with GFP.

### 4.10. Statistics

All data are expressed as mean ± SEM, and statistics were calculated using the GraphPad Prism 8.0 software. In experiments with normal distribution, statistical analyses were performed using the unpaired two-sided Student’s *t* test, one-way ANOVA and two-way ANOVA with Bonferroni’s or Tukey’s post hoc tests. T test with Welch’s correction was applied when variances were unequal. Values of *p* < 0.05 were considered statistically significant. Grubbs and ROUT tests were performed to determine the significant outlier values.

## 5. Conclusions

In conclusion, our study points out the hippocampal CA1 as a brain region sensitive to the duration of chronic stress and that Egr1 in superficial CA1 pyramidal neurons may play a crucial role in stress-induced sequelae such as behavioral despair and impaired spatial learning. Finally, future studies should focus on the identification of Egr1 upstream and downstream molecular mechanisms underlying stress-induced sequelae.

## Figures and Tables

**Figure 1 ijms-24-03833-f001:**
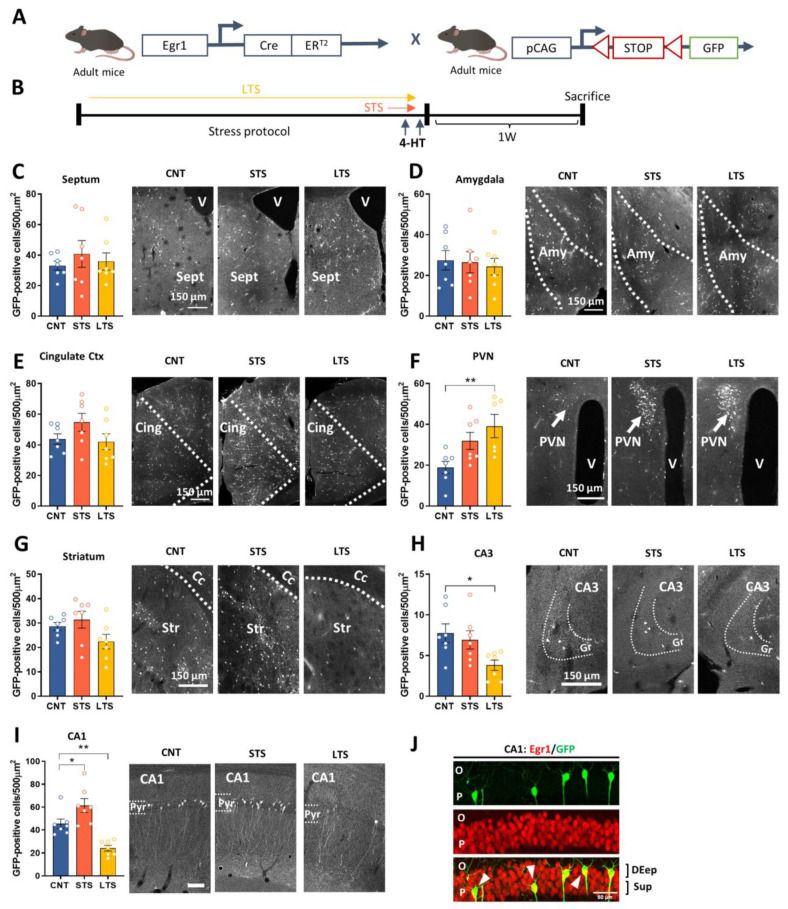
Egr1-dependent activated neuronal ensembles in different brain regions by chronic stress. (**A**), Schematic representation of double-heterozygous-mutant Egr1-CreERT2 × R26RCE GFP mice. (**B**), Egr1-CreERT2 × R26RCE mice were subjected to 0 (CNT), 2 (STS) or 28 (LTS) days of CUMS. For the last two days of the protocol, all mice received 50 mg/kg of 4-hydroxytamoxifen (4-HT, i.p), and one week later they were sacrificed. Representative images and quantification of Egr1-dependent activated cells (estimated number of GFP-positive cells/area of 500 µm^2^) per region in: the septum (**C**), one-way ANOVA group effect: F_(2,18)_ = 0.3858, *p* = 0.6854; the amygdala (**D**), group effect: F_(2,18)_ = 0.1160, *p* = 0.8911; the cingulate cortex (**E**), group effect: F_(2,18)_ = 1.946, *p* = 0.1718; the paraventricular nucleus (PVN) (**F**), group effect: F_(2,17)_ = 5.686, *p* = 0.0129; the striatum (**G**), group effect: F_(2,18)_ = 2.716, *p* = 0.0931; the CA3 (**H**), group effect: F_(2,18)_ = 4.322, *p* = 0.0293, and the CA1 (**I**), group effect: F_(2,18)_ = 17.63, *p* < 0.0001; Scale bar, 50 µm. The values are expressed as the mean ± SEM, N = 7 adult mice per condition. Dunnet’s post hoc test, * *p* < 0.05, ** *p* < 0.005 compared with CNT. (**J**) Representative images of the CA1 of the hippocampus showing GFP-activated cells (green) and Egr1-positive cells (red). White arrows indicate double-Egr1-positive and GFP-positive cells. V, ventricle; CC, corpus callosum; O, Stratum oriens; P, Stratum pyramidale.

**Figure 2 ijms-24-03833-f002:**
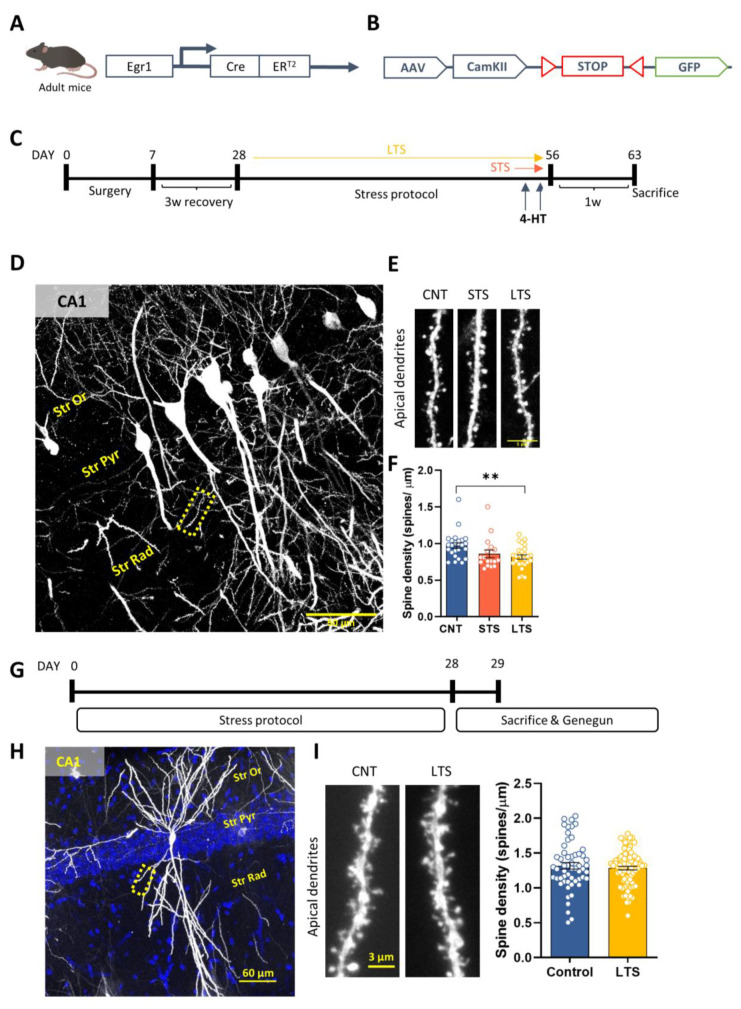
CUMS specifically alters dendritic spine density in Egr1-activated pyramidal neurons. (**A**), Schematic representation of the Egr1-CreERT2 mouse model and (**B**) AAV expressing GFP under the control of a STOP floxed codon. (**C**), Timeline of the experiment: Adult mice were injected with AAV-CaMKII-Flex-GFP. After recovery, they were exposed to 0 (CNT) or 2 (STS, orange arrow) or 28 days of CUMS (LTS, yellow arrow). On the last two days of CUMS, all animals received 50 mg/kg of 4-HT (i.p.) 30 min before the stressor. One week after the last stressor, animals were sacrificed for tissue collection. (**D**), Representative image showing labelled neurons in the CA1 region of STS mice. Effect of CUMS on apical spine density (**E**,**F**). In (**F**), N = 17–27 dendrites from seven different mice per group were quantified. The values are means ± SEM. Statistical analysis with one-way ANOVA followed by Tukey’s post hoc test, (group effect in (**F**), F_(2, 47)_ = 5.86, *p* = 0.005). ** *p* < 0.005 compared with CNT group. (**G**), WT mice were subjected to 0 (CNT) or 28 days (LTS) of CUMS. Forty-eight hours after the last stressor, mice were perfused and hippocampal slices subjected to the gene gun method. (**H**), Representative image showing a labelled neuron of the CA1. (**I**), Representative image (left panel) and quantification (right panel) showing no effect of CUMS in apical dendrites of CA1 pyramidal cells in LTS mice compared with CNT mice (unpaired *t*-test, t_142_ = 0.6730, *p*= 0.5020). In (**H**), 58–88 dendrites from seven different mice per group were quantified.

**Figure 3 ijms-24-03833-f003:**
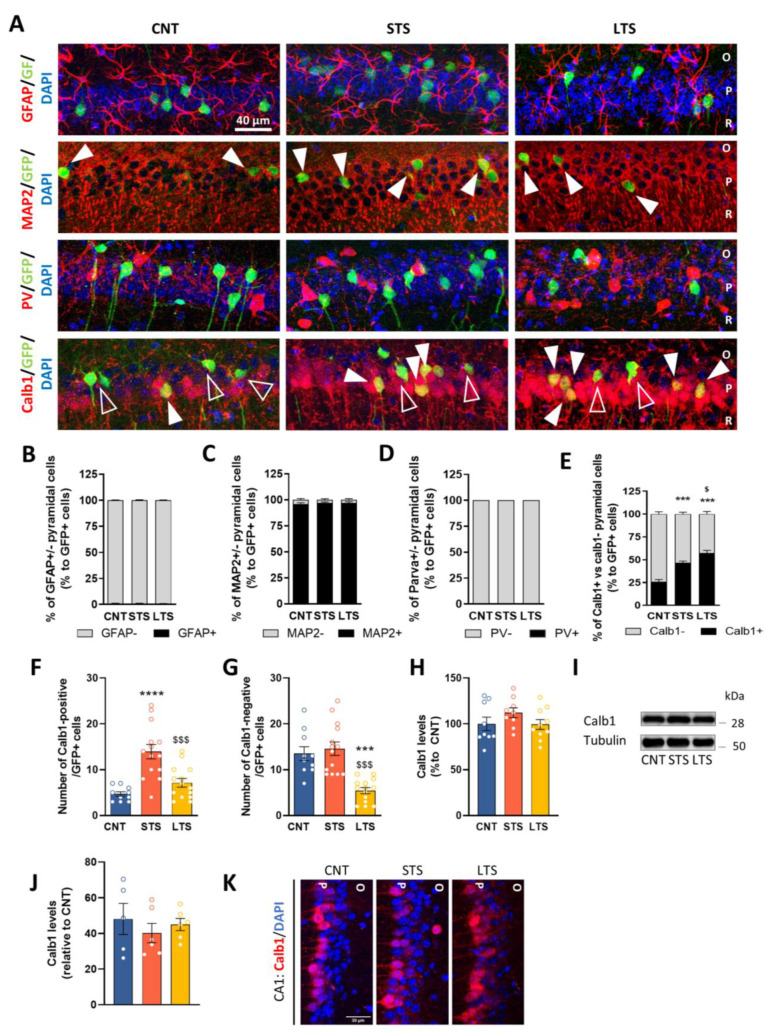
Neural identity characterization of Egr1-dependent activated cells by short- or long-term stress in the CA1. (**A**), Representative images showing the identity of the activated cells (green) by co-labeling with specific cells markers (red). In all rows, GFP-positive cells are in green. First row: representative images from all conditions (CNT, STS and LTS) showing labeling of astrocytes (red). Second row: labeling of MAP2 (red). White arrows point to GFP- and MAP2-double-positive neurons. Third row: labeling of parvalbumin (PV, red). Fourth row: labeling of calbindin1 (Calb1, red). White arrows point to double-positive GFP- and calbindin-double-positive neurons. Open arrows point to GFP-positive calbindin-negative cells. (**B**–**E**), Quantification of the percentages of double-positive cells in CA1 for GFP and GFAP (**B**), GFP and MAP2 (**C**), GFP and PV (**D**), and GFP and Calb1 (**E**). Values are means ± SEM, N = 7 mice per group. In (**E**), statistical analysis with one-way ANOVA, group effect: F_(2,18)_ = 42.31, *p* < 0.0001. Total number of GFP-positive cells per surface unit that are also calbindin1-positive (**F**) or calbindin-negative (**G**). Statistical analysis with one way ANOVA, in (**F**), F_(2,35)_ = 15.91, *p* < 0.0001 and in (**G**), F_(2,35)_ = 17.40, *p* < 0.0001. Densitometry quantification (**H**) and representative immunoblots (**I**) of calbindin levels in the hippocampal lysates of CNT, STS and LTS mice. N = 9–10 mice per group. Quantification of calbindin intensity in the CA1 (**J**) and representative images (**K**) of CNT, STS and LTS mice, N= 5–6 mice per group. All values are expressed as means ± SEM. Tukey’s post hoc test, **** *p* < 0.0001 or *** *p* < 0.001 compared to CNT and $ < 0.05, $$$ *p* < 0.001 compared to STS mice. O, Stratum oriens; R, Stratum radiatum; P, Stratum pyramidale.

**Figure 4 ijms-24-03833-f004:**
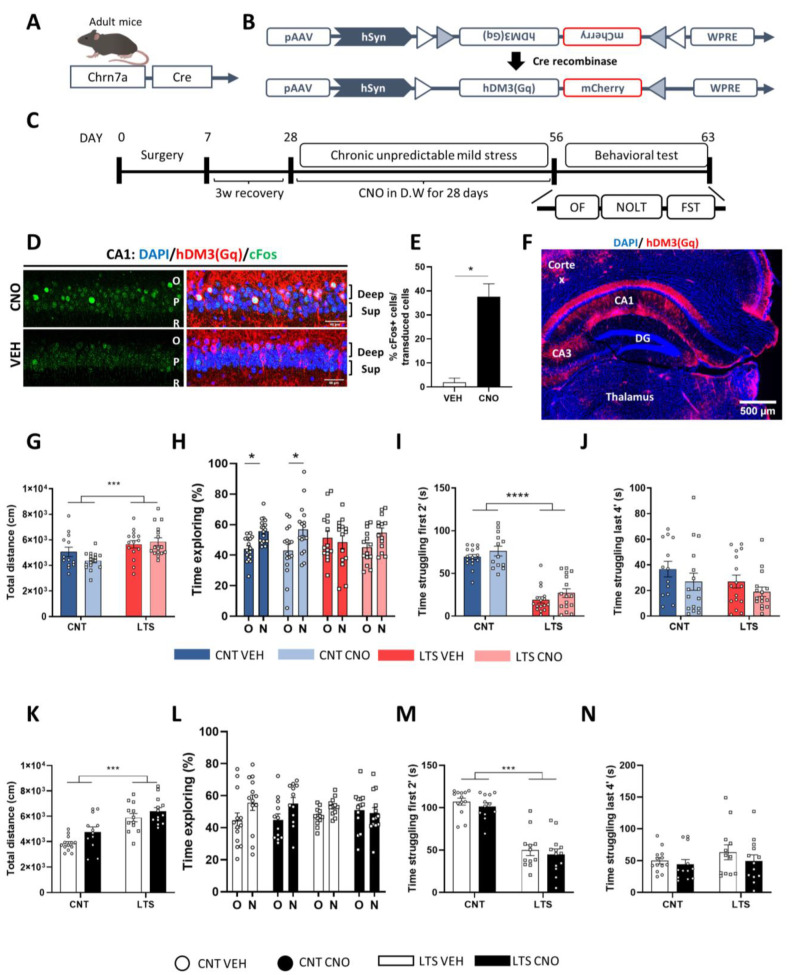
Effects of the chemogenetic activation of CA1 deep pyramidal neurons during CUMS. (**A**), Schematic illustration of the Chrn7a-Cre mouse model and of the AAV (**B**) used to express the activator DREADD (hDM3(Gq)) after Cre recombination. (**C**), Timeline of the experiment. Mice were injected with the vector shown in (**B**) and, after recovery, they were subjected to the CUMS protocol. During the stress protocol, mice received either CNO (1 mg/kg) or vehicle (VEH) solution in drinking water. Twenty-four h after the last stressor, mice were subjected to a battery of behavioral tests. (**D**), Representative image of CA1 showing cFos-positive cells (green) and viral transduction (mCherry, red) and cell nuclei (blue, DAPI). (**E**), Percentage of cFos+ cells (green) co-labelled with mCherry (red) in mice treated with VEH or CNO (unpaired *t*-test, t_17_ = 2.83, *p* = 0.01). A representative global image (**F**) from (**D**) of viral transduction specifically in the hippocampal CA1. In the open field, locomotor activity (**G**) was monitored for 15 min. Two-way ANOVA, stress effect: F_(1,56)_ = 13.23, *p* = 0.0006. (**H**), In the novel object location test, spatial memory was evaluated 24 h after a training trial as the percentage of total time spent exploring either the object placed at a new location (**N**) or the object placed at the old location (**O**). Two-way ANOVA, new location effect: F_(1,122)_ = 8.64, *p* = 0.0039, N = 13–18 mice per condition. In the forced-swim test, the immobility time was evaluated during the first 2 min (**I**) and the last 4 min (**J**) of the 6 min trial in all groups. In the first 2 min, two-way ANOVA, group effect: F_(1,57)_ = 128.6, *p* < 0.0001. An independent new cohort of WT mice underwent CUMS or not and were treated with vehicle or CNO to determine undesired effects of CNO. We used the same experimental design as in **c** without surgery. (**K**), In the open field, locomotor activity was monitored for 15 min. Two-way ANOVA, stress effect: F_(1,46)_ = 37.43, *p* < 0.0001. (**L**), Spatial memory was evaluated in the novel object location test. Two-way ANOVA, new location effect: F_(1,92)_ = 6.1, *p*= 0.0152. In the forced-swim test, the immobility time was evaluated during the first 2 min (**M**) and the last 4 min (**N**) of the 6 min trial in all groups. In the first 2 min, two-way ANOVA, group effect, F_(1,46)_ = 113, *p* < 0.0001. N= 12–13 mice per condition. All values are means ± SEM. Tukey’s post hoc in (**G**,**I**–**K**,**M**); Bonferroni post hoc in (**H**,**L**). * *p* < 0.05, *** *p* < 0.001 and **** *p* < 0.0001. O, Stratum oriens; R, Stratum radiatum; P, Stratum pyramidale.

**Figure 5 ijms-24-03833-f005:**
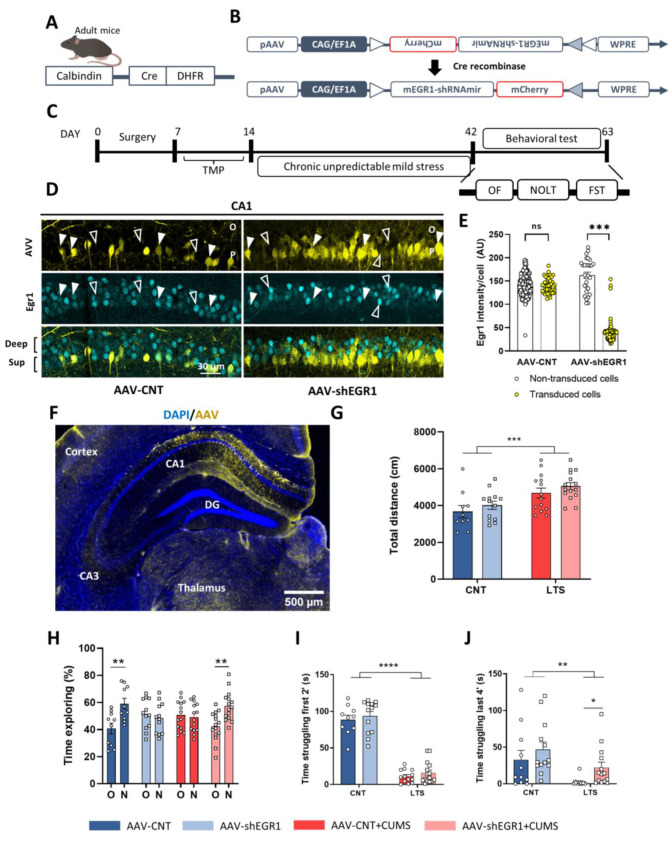
Effects of Egr1 shRNA delivery in CA1 superficial pyramidal neurons during CUMS. (**A**), Schematic representation of the calbindin-Cre mice. (**B**), Schematic diagram showing the AAV-mCherry-floxed-shEGR1 viral construct. (**C**), Timeline of the experiment. Mice were injected with the construct shown in (**B**) or with a control floxed construct and, after surgery, they received TMP to induce Cre expression. Three weeks after surgery, they were subjected to the CUMS protocol or stayed in their home cage. (**D**), Representative images in CA1 showing Egr1 levels (cyan) in mice injected with control floxed virus (AAV-CNT, left panel) or with the AAV containing the floxed shRNA against Egr1 (AAV-shEGR1). Viral expression is depicted in yellow. White arrows designate transduced neurons expressing the virus. Open arrows designate non-transduced neurons. (**E**), Quantification of Egr1 optical density (IOD) in non-transduced cells (white dots) and in transduced cells with the AAV (yellow dots) in each condition (AAV-CNT or AAV-shEGR1). T-test: t_167_ = 56.77, *** *p* < 0.001 (ns, not significant). A representative global image (**F**), from (**D**) of viral transduction specifically in the hippocampal CA1. In the open field, locomotor activity (**G**) was monitored for 15 min. Locomotor activity, two-way ANOVA, group effect: F_(1,48)_ = 16.36, *p* < 0.0001. In the novel object location test (**H**), spatial memory was evaluated 24 h after a training trial as percentage of time exploring the object placed in a new location (N) versus the time exploring the object placed in an old location (O). Two-way ANOVA, object in a new location effect: F_(1,94)_ =10.33, *p* = 0.0018. Interaction effect: F_(3,95)_= 5.878, *p* = 0.001. In the forced-swim test, the immobility time was evaluated during the first 2 min (**I**) and the last 4 min (**J**) of the 6 min trial in all groups. In (**I**), two-way ANOVA, group effect: F_(1, 47)_ = 257.7, *p* < 0.0001. In **J**, two-way ANOVA, group effect: F_(1 47)_ = 10.28, *p* = 0.0024. T-test between LTS mice (t_25_ = 2.568, *p* = 0.0166). N= 12–13 mice per condition. All data is represented as mean ± SEM. Tukey’s was used as a post hoc * *p* < 0.05, ** *p* < 0.005, **** *p* < 0.001. O, Stratum oriens; P, Stratum pyramidale.

**Table 1 ijms-24-03833-t001:** Schematic organization of the chronic unpredictable mild stress protocol.

		GROUP		
TIME	STRESSOR	CNT	STS	LTS
Day 0	Isolation			Isolation
Day 1	Home cage inclination (1 h)			
Day 2	Food deprivation (24 h)			
Day 3	Restraint (1 h)			
Day 4	Water deprivation (24 h)			
Day 5	Light–dark cycle alterations (24 h)			
Day 6	Forced Swim (5 min)			
Day 7	Exposition to rat sawdust (4 h)			
Day 8	Home cage inclination (1 h)			
Day 9	Water deprivation (24 h)			
Day 10	Restraint (1 h)			
Day 11	Light–dark cycle alterations (24 h)			
Day 12	Food deprivation (24 h)			
Day 13	Restraint (1 h)			
Day 14	Exposition to rat sawdust (4 h)			
Day 15	Home cage inclination (1 h)			
Day 16	Light–dark cycle alterations (24 h)			
Day 17	Forced Swim (5 min)			
Day 18	Exposition to rat sawdust (4 h)			
Day 19	Water deprivation (24 h)			
Day 20	Forced Swim (5 min)			
Day 21	Exposition to rat sawdust (4 h)			
Day 22	Home cage inclination (1 h)			
Day 23	Food deprivation (24 h)			
Day 24	Water deprivation (24 h)			
Day 25	Light–dark cycle alterations (24 h)			
Day 26	Food deprivation (24 h)		Isolation	
Day 27	Restraint (1 h)	+4-HT	+4-HT	+4-HT
Day 28	Forced Swim (5 min)	+4-HT	+4-HT	+4-HT

Left column: day of stress; second column: stressor type; third column: control non-stressed group of mice; fourth column: short-term-stressed group of mice (2 days of stress); fifth column: long-term-stressed group of mice (28 days of stress). Note that gray cells illustrate real days in which each group of mice received a stressful stimulus. 4-HT: 4-Hydroxytamoxifen.

## Data Availability

The datasets generated during and/or analyzed during the current study are available from the corresponding author on reasonable request.

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
