# Peer review of "Cognitive and Emotional Symptoms Induced by Chronic Stress Are Regulated by EGR1 in a Subpopulation of Hippocampal Pyramidal Neurons"

_ijms, 2023, doi:10.3390/ijms24043833_

Round 1
Reviewer 1 Report
Chronic stress, a major environmental risk for several neuropsychiatric disorders, functionally impact the hippocampus, a brain region critically involved in memory. IEGs are early response genes that are dynamically regulated in response to neuronal activity and neuronal insults. They have been widely used as a marker for neuronal populations that undergo plastic changes underlying memory. Egr1 is an IEG involved in synaptic plasticity, but its role in stress effects is poorly understood. In this study, the authors characterized stress-induced Egr1+ neuronal ensembles and found changes in different brain regions including the hippocampus. They found that in CA1, Egr1 expression is sensitive to stress duration. They claim that Egr1 activation is differentially regulated in deep vs superficial CA1 pyramidal cells by chronic vs short-term stress. In addition, removing Egr1 from superficial CA1 pyramidal cells impacted stress-induced behavioral phenotypes while activation of deep CA1 neurons did not impact stress effects on behavior. Experiments are well designed and supported by number of controls. In general, conclusions are supported by experimental evidence. However, some points need to be addressed in order to fully support all of the claims.
MAJOR POINTS:
1. In figure 3, The authors used immunostaining for Calb1 to identifies the superficial pyramidal neurons of CA1. They claim that there is a transient increase in the percentage of Egr1-dependent activated neurons in superficial pyramidal neurons and that deep pyramidal neurons are severely and progressively deactivated (in terms of Egr1-dependent activation) at the end of the CUMS protocol. It is important to test whether the total number and the distribution of calb1 are altered by the CUMS protocol. If Calb1 expression is altered by CUMS, the authors should also assess deep vs superficial Egr1+ CA1 neuronal ensembles using location in the pyramidal layer, independently of Calb1 staining. This point is also important for the experiment using Calb-cre mice to reduce EGR1 expression selectively in CA1 superficial pyramidal neurons
2. The authors should provide more information about the control AAV used for the EGR1-shRNA experiments. AAV5-CAG/EF1a-DIO-mCherry-mEGR1-shRNAmir was used to reduce EGR1 expression but what was the control virus? Was AAV5-CAG/EF1a-DIO-mCherry-mEGR1-shRNAmir generated for this paper? The authors should cite the paper generating this AAV or indicate the shRNA sequence (if generated in this study).
MINOR POINTS:
Rationale for using Chrna7-Cre and Calb1-Cre is a bit missing in the abstract. In the abstract the authors should also mention why they decide to investigate IEG in stress.
Author Response
Professor Milica Drenovac
Guest editor
International Journal of Molecular Sciences
Barcelona, February 07th, 2023
Dear professor Milica Drenovac
We would like to re-submit our manuscript entitled "Cognitive and Emotional Symptoms Induced by Chronic Stress Are Regulated by EGR1 in a Subpopulation of Hippocampal Pyramidal Neurons" by Sancho-Balsells et al. Our work is original research, it has not been previously published and it has not been submitted for publication elsewhere while under consideration. All authors declare there is no conflict of interest.
We thank the reviewers for their thoughtful comments, which we believe have improved the quality of the revised manuscript. We have addressed all reviewer’s concerns by including new experimental data as well as text amendments. Our responses are enclosed below each reviewer’s comment.
Please note that the modified text in the manuscript is marked up in red.
MAJOR POINTS:
- In figure 3, The authors used immunostaining for Calb1 to identifies the superficial pyramidal neurons of CA1. They claim that there is a transient increase in the percentage of Egr1-dependent activated neurons in superficial pyramidal neurons and that deep pyramidal neurons are severely and progressively deactivated (in terms of Egr1-dependent activation) at the end of the CUMS protocol. It is important to test whether the total number and the distribution of calb1 are altered by the CUMS protocol. If Calb1 expression is altered by CUMS, the authors should also assess deep vs superficial Egr1+ CA1 neuronal ensembles using location in the pyramidal layer, independently of Calb1 staining. This point is also important for the experiment using Calb-cre mice to reduce EGR1 expression selectively in CA1 superficial pyramidal neurons.
We agree with the reviewer that this missing information is essential for the general conclusions of the manuscript. To analyze whether calbindin levels are altered by the CUMS protocol we performed two additional experiments. First, we used Western Blot to measure total levels of Calbindin in mice that underwent 2 days (STS), 28 days (LTS) or no stress (CNT). Results showed that there are no significant differences between conditions (Figure 3H and 3I). Moreover, we performed immunostaining in those same mice to gain precision in the data obtained. We observed no changes in Calbindin intensity optical density (IOD) in the CA1 pyramidals between CNT, STS, and LTS mice (Figure 3J and 3K).
Still using immunofluorescence, we then analyzed the levels of calbindin in those Calb-cre mice shown in Figure 5. We again didn’t find any change between the groups analyzed (CNT GFP, CNT SH, LTS GFP, and LTS SH) (See Supplementary Figure 1).
We have added this new information in the material and methods section (See Sections 4.7. and 4.9.). We have also modified the correspondent results sections and the figure legend.
- The authors should provide more information about the control AAV used for the EGR1-shRNA experiments. AAV5-CAG/EF1a-DIO-mCherry-mEGR1-shRNAmir was used to reduce EGR1 expression but what was the control virus? Was AAV5-CAG/EF1a-DIO-mCherry-mEGR1-shRNAmir generated for this paper? The authors should cite the paper generating this AAV or indicate the shRNA sequence (if generated in this study).
We agree with the reviewer that this information is necessary. We have added this information in the section “4.3. Stereotaxic surgery and viral transduction in vivo”, line 476. For the Egr1-shRNA experiments, we used an AAV5-CAG-FLEX-GFP virus as a control. We agree that a better control virus would be an AAV5-CAG/EF1a-DIO-mCherry-mEGR1-shRNAmir. Since this could be a limitation of the study, we have added in the text the following sentence in lines 303-305: “It is noteworthy that since in this experiment we used as a control the AAV5-CAG-FLEX-GFP instead of a scramble sequence they should be taken with caution”.
We have added this information in the pertinent result section in lines 276-279: “We thus transduced only superficial CA1 pyramidal (Calb1-positive) with shRNA against Egr1 (Figure 5B) or with a GFP flex CNT virus”
MINOR POINTS:
Rationale for using Chrna7-Cre and Calb1-Cre is a bit missing in the abstract. In the abstract the authors should also mention why they decide to investigate IEG in stress.
We agree with the reviewer that the rationale for using these mice might not be clear enough. We have modified the abstract to make it clearer. See lines 30-33:
“Then, to specifically manipulate deep and superficial pyramidal neurons of the hippocampus, we used the Chrna7-Cre (to express Cre in deep neurons) and Calb1-Cre mice (to express Cre in superficial neurons).”
Moreover, we used these two markers because they have been previously described as appropriate to classify hippocampal subpopulations. Using the hippocampus RNA-Seq atlas (doi: 10.7554/eLife.14997) we found that Chrna7 is enriched in deep neurons whereas Calb1 is enriched in superficial neurons. We also found different papers stating that these two hippocampal subpopulations can be classified using these two makers (doi:10.1038/nn.4074, https://doi.org/10.1016/j.conb.2018.04.013; https://doi-org.sire.ub.edu/10.1038/s41593-018-0118-0)
Regarding to the second point, we apologize for the misunderstanding, but we did not decide to investigate IEG in stress. Instead, we used first the Egr1-CreERT2 x R26RCE mouse model as a tool to map activation of neural ensembles in different phases of chronic stress (See lines 90-96 for rationale). Secondly, since we observed potential relevant results using these tool as a marker of neuronal activity, we then tested if Egr1 had a biological relevance per se as a potential underlying molecular mechanism (Figure 5).

Reviewer 2 Report
This manuscript tittled: “ Cognitive and Emotional Symptoms Induced by Chronic Stress Are Regulated by EGR1 in a Subpopulation of Hippocampal Pyramidal Neurons” describe in an interesting way the role of Egr1 expression in the superficial CA1 pyramidal neurons in stress conditions. However, there are some important points should be considered and corrected, please attend the following comments.
Mayor Comments:
· In the design experiment of figure 2A, you mention the protocol was performed at 28PD (prepuber age) and it was concluded at 56 PD (postpuber age), so your protocol was made in the neurodevelopmental stage where different events are presents like pruning and neurogenesis, I think it´s important to discuss the role of this kind population of Hippocampal Neurons with the neurodevelopmental process, I mean not only considering the stress condition.
· In figures 4L and 5 H you presented results from Novel Object Recognition Test, but I recommend calculate discrimination index to show clearly the effects of each experimental conditions, this provide a better perspective (see for review 10.1016/j.jchemneu.2021.102057)
Minor Comments:
· In materials and methods section please specify the age of the animals and the numbers of animals for each group that were used in each experiments.
· It´s important include an explanation with more details about the counting method you used for the Spine Density analysis. ¿How many micrometers did you considered for the counting of spines? Because you present in the graphs the number per µm, but in some morphological studies the spines density analysis is performed counting per each 10 µm. Even more, did you considered distal or proximal dendrites? for the quantification of the spines.
Author Response
Professor Milica Drenovac
Guest editor
International Journal of Molecular Sciences
Barcelona, February 07th, 2023
Dear professor Milica Drenovac
We would like to re-submit our manuscript entitled "Cognitive and Emotional Symptoms Induced by Chronic Stress Are Regulated by EGR1 in a Subpopulation of Hippocampal Pyramidal Neurons" by Sancho-Balsells et al. Our work is original research, it has not been previously published and it has not been submitted for publication elsewhere while under consideration. All authors declare there is no conflict of interest.
We thank the reviewers for their thoughtful comments, which we believe have improved the quality of the revised manuscript. We have addressed all reviewer’s concerns by including new experimental data as well as text amendments. Our responses are enclosed below each reviewer’s comment.
Please note that the modified text in the manuscript is marked up in red.
Reviewer #2
This manuscript titled: “Cognitive and Emotional Symptoms Induced by Chronic Stress Are Regulated by EGR1 in a Subpopulation of Hippocampal Pyramidal Neurons” describe in an interesting way the role of Egr1 expression in the superficial CA1 pyramidal neurons in stress conditions. However, there are some important points should be considered and corrected, please attend the following comments.
Major Comments:
In the design experiment of figure 2A, you mention the protocol was performed at 28PD (prepuber age) and it was concluded at 56 PD (postpuber age), so your protocol was made in the neurodevelopmental stage where different events are presents like pruning and neurogenesis, I think it´s important to discuss the role of this kind population of Hippocampal Neurons with the neurodevelopmental process, I mean not only considering the stress condition.
We apologize for the misunderstanding. We did NOT use or performed any protocol at prepuber ages. In contrast, in the entire manuscript, we always applied the stress protocol to adult mice. In Figure 2A, adult mice received 28 days of stress. Here, the Day 0 is considered de first day of stress and the Day 28, the last day of the stress protocol. In order to address this point we have added the term “adult mice” in Figures 1A, 2A, 4A and 5A. We thank the reviewer for this appreciation, and we hope that now everything is clear. Moreover, we have added a new paragraph in the Material and Methods Section 4.1. “Animals” stating the number and age of the animals used in each experiment.
In figures 4L and 5 H you presented results from Novel Object Recognition Test, but I recommend calculate discrimination index to show clearly the effects of each experimental conditions, this provide a better perspective (see for review 10.1016/j.jchemneu.2021.102057)
We agree with the reviewer that the discrimination index is an additional and useful parameter to show the effects of the experimental conditions. We have applied this index to our results in figure 5 (see below). As the reviewer can observe, the trend is very visual but not significant. Therefore, we suggest showing this result to the reviewer only for his/her commodity but to maintain the original analysis since it examines whether each individual group actually discriminates between the novel and the familiar object, also because it is recommended (Akkerman et al., 2012; Behav Brain Res. Doi: 10.1016/j.bbr.2012.03.024) and finally because it is widely employed:
- Singh P, Thakur MK. Reduced recognition memory is correlated with decrease in DNA methyltransferase1 and increase in histone deacetylase2 protein expression in old male mice. Biogerontology. 2014 Aug;15(4):339-46.
- Kushwaha A, Thakur MK. Increase in hippocampal histone H3K9me3 is negatively correlated with memory in old male mice. Biogerontology. 2020 Apr;21(2):175-189.
- Landreth K, Simanaviciute U, Fletcher J, Grayson B, Grant RA, Harte MH, Gigg J. Dissociating the effects of distraction and proactive interference on object memory through tests of novelty preference. Brain Neurosci Adv. 2021 Apr 27;5:23982128211003199.
- Mastrolia V, Al Massadi O, de Pins B, Girault JA. Pyk2 in dorsal hippocampus plays a selective role in spatial memory and synaptic plasticity. Sci Rep. 2021 Aug 11;11(1):16357.
- Giralt A, Brito V, Chevy Q, Simonnet C, Otsu Y, Cifuentes-Díaz C, de Pins B, Coura R, Alberch J, Ginés S, Poncer JC, Girault JA. Pyk2 modulates hippocampal excitatory synapses and contributes to cognitive deficits in a Huntington's disease model. Nat Commun. 2017 May 30;8:15592.
- Muñoz-Castañeda R, Díaz D, Peris L, Andrieux A, Bosc C, Muñoz-Castañeda JM, Janke C, Alonso JR, Moutin MJ, Weruaga E. Cytoskeleton stability is essential for the integrity of the cerebellum and its motor- and affective-related behaviors. Sci Rep. 2018 Feb 15;8(1):3072.
- Costa G, Serra M, Simola N. Association between Novel Object Recognition/Spontaneous Alternation Behavior and Emission of Ultrasonic Vocalizations in Rats: Possible Relevance to the Study of Memory. Brain Sci. 2021 Aug 9;11(8):1053.
- Brito V, Giralt A, Enriquez-Barreto L, Puigdellívol M, Suelves N, Zamora-Moratalla A, Ballesteros JJ, Martín ED, Dominguez-Iturza N, Morales M, Alberch J, Ginés S. Neurotrophin receptor p75(NTR) mediates Huntington's disease-associated synaptic and memory dysfunction. J Clin Invest. 2014 Oct;124(10):4411-28.
PASTED GRAPH (see attached document)
Minor Comments:
In materials and methods section please specify the age of the animals and the numbers of animals for each group that were used in each experiment.
We have added this information in the Section “4.1. Animals”, lines 454-464.
“For the experiments shown in Figure 1 and 3, 7 adult (4-month-old) male mice were used per group (total n=21 mice). For the experiments shown in Figure 2A-2F, 7 adult (6-7-month-old) male mice were used per group (total n= 21 mice). For the experiments shown in Figure 2G-2I, 3-4-month-old mice were employed. For the experiments shown in Figure 4A-4G, 4I and 4J, 60 adult (5-6-month-old) mice were used (CNT VEH n= 13; CNT CNO n=16; LTS VEH n=15 and LTS CNO n=16). In figure 4H, adult (5-6 month-old) mice were used (CNT VEH=18; CNT CNO n=16; LTS VEH n=15 and LTS CNO n=16). For the experiments shown in Figure 4K-N, 50 male adult mice (5–6-month-old) were used (CNT VEH n=12; CNT CNO n=12; LTS VEH n=13 and LTS CNO n= 13). For the experiments shown in Figure 5, 52 male adult mice (7-8-month-old) were used (CNT GFP n=10; CNT SH n=13; LTS GFP n=14 and LTS SH n=15).”
It´s important include an explanation with more details about the counting method you used for the Spine Density analysis. ¿How many micrometers did you considered for the counting of spines? Because you present in the graphs the number per µm, but in some morphological studies the spines density analysis is performed counting per each 10 µm. Even more, did you consider distal or proximal dendrites? for the quantification of the spines.
We thank the reviewer for this suggestion. We have now added more details about spine density analysis in the Materials and methods, section “Spine density analysis”, lines 588-591.
“For spine density analysis, we examine second-order dendrites. Thus, ramifications from the apical dendrite were analyzed. On average, 30um ± 2 of length were evaluated per dendrite and the ratio was performed as the number of dendrites/total length analyzed. 17-27 dendrites from 7 different mice per group were quantified.”

Round 2
Reviewer 1 Report
The authors addressed all my concerns. I do not have any further request.